# Every Single Specimen Counts: A New Docosia Winnertz (Diptera: Mycetophilidae) Species Described from a Singleton [note 1]

**DOI:** 10.3390/insects12121069

**Published:** 2021-11-29

**Authors:** Olavi Kurina, Heli Kirik

**Affiliations:** Institute of Agricultural and Environmental Sciences, Estonian University of Life Sciences, Friedrich Reinhold Kreutzwaldi 5-D, 51006 Tartu, Estonia; heli.kirik@emu.ee

**Keywords:** COI, fungus gnats, Georgia, new species, singleton, taxonomy

## Abstract

**Simple Summary:**

A new fungus gnat species has been described from a single specimen collected from Georgia (Sakartvelo). The new species, named after its occurrence in Caucasia as *Docosia caucasica* sp. n., is distinguished from congeners by the characters in male terminalia and a unique COI sequence. As a substantial proportion of species in ecological communities tend to be rare, about 20–30% of new insect taxa have been described from a singleton so far. Therefore, following high-quality standards when describing new species, particularly when dealing with minimalistic material, is crucial. As much as possible, using multiple sets of characters, like morphology and DNA sequencing, is encouraged.

**Abstract:**

A new species—*Docosia caucasica* sp. n.—has been described from material collected from the Lesser Caucasus Mountains in Georgia (Sakartvelo). The new species belongs to a group of Palaearctic species characterized by distinct posterolateral processes of gonocoxites and apically modified setae at the posteroventral margin of the gonocoxites medially. Within the group, *D. caucasica* sp. n. is most similar to *D. landrocki* Laštovka and Ševčík, 2006 in having a similar outline of the medial process of posteroventral margin of the gonocoxites and the gonostylus. There is also a marked difference within the partial cytochrome c oxidase subunit 1 gene (COI) sequence of *D. caucasica* sp. n. and other *Docosia* spp. available in public databases. As the new species is described from a single male specimen only, the adequacy and code compliance of that are discussed.

## 1. Introduction

The genus *Docosia* Winnertz, 1864 of Mycetophilidae (Diptera) has gained considerable attention during the last few decades, based on both taxonomy as well as phylogeny. Having been classified traditionally within the subfamily Leiinae (or the tribus Leiini) by earlier authors [1,2,3], molecular studies have instead suggested it belongs to Gnoristinae [4,5]. However, in a recent synopsis of Leiinae by Oliveira and Amorim [6], a new subfamily Tetragoneurinae was established, consisting of four genera including *Docosia*. Intrageneric phylogeny based on five molecular markers has been proposed by Ševčík et al. [7]. Supplementing phylogenetic studies, a number of new species have been described during the last few decades from the Palaearctic (e.g. [8,9,10,11,12,13]), Nearctic [14,15], Oriental [16] and Neotropical [17,18] realms. Altogether, 86 extant *Docosia* species have been described, including 62 from the Palaearctic realm [7]. Within Mycetophilidae, *Docosia* species are characterised by medium size, dark brown to black body colour, unmarked wings, details of wing venation and distinctive male terminalia with combs of small spines (retinacula) on cerci in particular [12,19]. The species level identification has so far relied mainly on the morphology of male terminalia. However, a considerable number of species has also DNA information uploaded to the public databases (https://www.boldsystems.org/ (accessed on 2 November 2021); https://www.ncbi.nlm.nih.gov/genbank/ (accessed on 2 November 2021)) that enable and support species identification.

This communication is prompted by discovery of a new *Docosia* species from the material collected in Georgia (Sakartvelo). Because the new species is represented by a single male specimen only, the adequacy and code compliance of such a description is discussed.

## 2. Materials and Methods

The specimen was collected during 2019 in the Lesser Caucasus Mountains in Georgia by collaborators of ZFMK (Zoological Research Museum Alexander Koenig, Bonn, Germany) using a Malaise trap. The collecting locality was a small meadow (flat surface close to a small river) surrounded by riverine trees and forest. The material was collected into and thereafter preserved in ethyl alcohol. Fungus gnats were sorted out and identified, whereas one specimen was found to represent a new species and therefore subjected to detailed study.

One fore leg was detached and used for DNA sequencing. The sample was homogenized with Kontes Pellet Pestle (DWK Life Sciences GmbH, Mainz, Germany) and DNA extraction completed using DNeasy Blood and Tissue Kit (Qiagen, Hilden, Germany) according to manufacturer instructions. 710 bp partial cytochrome c oxidase subunit 1 (COI) sequences were replicated using the primer pair LCO1490 (5′- GGTCAACAAATCATAAAGATATTGG-3′) and HCO2198 (5′- TAAACTTCAGGGTGACCAAAAAAT-3′) [20], in a PCR mixture comprising 1 µL template DNA, 12.5 µL DreamTaq PCR Master Mix (Thermo Fisher Scientific, Waltham, MA, USA), 0.5 µL of each primer and 10.5 µL ddH2O. The PCR program included a 15 min initial denaturation stage at 94 °C, followed by 60 cycles of 30 s denaturation at 94 °C, 30 s annealing at 44 °C and 30 s of synthesis at 72 °C, finished by a 10 min syntheses stage at 72 °C. Sequencing was carried out with Sanger sequencing by the Institute of Genomics Core Facility (University of Tartu, Tartu, Estonia). Forward and reverse strands were integrated and primer sequences trimmed in BioEdit version 7.2.6.1 [21]. The resulting sequence was used to search the GenBank database (National Institutes of Health, Bethesda, USA) for closely related species. COI sequences of *D. anatolica* Ševčík in Ševčík et al., 2020, *D. diutina* Plassmann, 1996, *D. fumosa* Edwards, 1925, *D. landrocki* Laštovka and Ševčík, 2006, *D. nigra* Landrock, 1928, *D. pannonica* Laštovka and Ševčík, 2006, *D. rohaceki* Ševčík, 2006 were used for genetic analysis in MEGA11 [22]. The phylogenetic tree was constructed using the maximum likelihood method and the appropriate model for analysis was chosen with the Find Best DNA/Protein Models (ML) function in MEGA11. Furthermore, bootstrapping with 1000 iterations was employed to test phylogeny. However, values less than 75% were removed from the final figure for ease of viewing. All in all, the analysis involved eight sequences and a total of 658 nucleotide positions.

For a detailed study of the terminalia, they were detached and treated with ca. 10% warm potassium hydroxide (KOH), followed by neutralization with acetic acid (CH_3_COOH) and washing with distilled water. Terminalia were studied in glycerine (C_3_H_8_O_3_) and stored as glycerine preparations in small plastic vials attached to the rest of the specimen [23]. Illustrations of the terminalia were prepared using a U-DA drawing tube attached to a compound microscope Olympus CX31. The digital images of the habitus and terminalia were combined using the software LAS V.4.1.0. from multiple gradually focused images taken by a Leica DFC 450 camera attached to a Leica 205C stereomicroscope. Adobe Photoshop CS5 and Topaz Shapen Al were used for editing the figures, compiling the plates and improving sharpness. Morphological terminology follows Søli [19] and Ševčík et al. [7]. 

The holotype of the new species is deposited in the Zoological Research Museum Alexander Koenig, Bonn, Germany (ZFMK). Comparative material of *Docosia landrocki* and other species are deposited in the Institute of Agricultural and Environmental Sciences, Estonian University of Life Sciences, Tartu, Estonia (IZBE).

## 3. Results

### 3.1. Description of New Species

***Docosia caucasica* sp. n.** (Figure 1 and Figure 2).

LSID urn:lsid:zoobank.org:act: 6B1E8E5E-E745-44B4-A270-24973EF6C913.

*Type material.* Holotype. Male, GEORGIA, Samtskhe-Javakheti, road from Abastumani to Saime, near river, 41°46.63′ N, 42°50.23′ E, 1370 m a.s.l. 10–11.vi.2019, Malaise trap, X. Mengual leg. (in ethyl alcohol, terminalia in glycerine, ZFMK-DIP-00082491; one fore leg used for DNA extraction, GenBank accession number: OL619794).

*Differential diagnosis.* Based on the structure of the male terminalia, *D. caucasica* sp. n. belongs to a group of Palaearctic species that have posteroventral margin of the gonocoxites with (1) a prominent medial process of variable shape bearing dense aggregation of modified setae, and (2) distinct lateral processes. In this group, the new species resembles *D. landrocki* Laštovka and Ševčík, 2006 by having a similar outline of the medial process of the posteroventral margin of the gonocoxites and the gonostylus [8] (Figure 8), (Figure 3A–F). However, *D. caucasica* sp. n. has (1) the posterolateral process of the gonocoxites ventrally tapering, extending to the posterior third of the gonostylus (posterolaterally drawn out to small process that bears an aggregation of medially directed long setae and an apical spine-like prolongation in *D. landrocki*), (2) dorsal lobe of the gonostylus posteromedially drawn out to a small extension (dorsal lobe of the gonostylus widening posteriorly, without an extension in *D. landrocki*), (3) ventral lobe of the gonostylus with posterior prong bearing two clearly delimited, apically pointed subequal spines (posterior prong with apically spathulate and basally fused two long subequal spines in *D. landrocki*), (4) ventral lobe of the gonostylus with anterior prong bearing an apical sabre-shaped spine (anterior prong apically bifid, with small spine apically on more posterior ramus in *D. landrocki*), (5) parameral apodeme in ventral view widened posteriorly (evenly rounded in *D. landrocki*), and (6) aedeagus with well elongated hook-like apical part (aedeagus hooked, not elongated in *D. landrocki*).

*Description*. Male (*n* = 1). Length of wing 2.7 mm; ratio of length to width 2.67.

Head blackish brown with numerous pale setae. Three ocelli, the laterals separated from the eye margins by approximately their own diameter. Mouthparts light brownish. Palpus with two basal segments brownish and three apical segments yellow. Scape, pedicel and all flagellomeres dark brown. Flagellomeres cylindrical, flagellomeres 1–13 about 1.6 times as long as broad, apical flagellomere conical, about 3.8 times as long as broad basally.

Thorax all dark brown to blackish but somewhat lighter than head, with light setae. Scutellum with numerous pale setae and several submarginal pale bristles about twice as long as scutellum. Antepronotum and proepisternum with pale bristles. Laterotergite and other pleural parts bare. Haltere pale yellow.

Legs. All coxae yellow, with basal third of mid- and hind coxae darkened. Trochanters brownish. Mid legs absent in holotype beyond trochanters. Fore- and hind femora mostly yellow, except hind femur apically darkened. Fore- and hind tibiae yellow. Fore tibia apicomedially with a semi-circular tibial organ (anteroapical depressed area), without strong setae, only densely covered with fine setulae. Tarsi seem darker because of dense setae.

Wing hyaline, unmarked. Radial veins and r-m dark brown, other veins paler, m-stem and basal parts of M_1_ and M_2_ faint, almost not traceable. Sc, Rs, bm-m, m-stem, basal part of M_4_ and basal half of cu-stem asetose, the other veins setose. Costa reaches to about the third of the distance between R_5_ and M_1_. Sc ends in R before the level of beginning of m-stem. Posterior fork begins before anterior fork, approximately at the middle of r-m.

Abdomen dark brown, with second and third segments lighter. Terminalia (Figure 2A–J) light brown. Tergite 9 basally rounded, with small medial incision, apically blunt, longer than broad, posterior margin slightly convex, setose with row of subapical stronger setae. Posteroventral margin of gonocoxites with extended flange; medial, somewhat tapering wide lobe densely covered with apically ramified setae. Dorsally from medial lobe and extending over it apically, spathulate, apically widened bare lobe, well discernible from posterior view. Laterally from medial lobe, posteroventral margin of gonocoxite drawn out to prominent posterolateral setose process extending to apical third of gonostylus. Gonostylus consists of two lobes: (1) large crescent-shaped dorsal lobe that is medially membranous, medially and laterally with short setae and with a long deviating subapical seta, and (2) ventral lobe that is split into three prongs, one medially directed anterior prong and two more posterior and posteriorly directed prongs, all with black apical spines. Parameral apodeme posteriorly widened and blunt. Adeagus with elongated hook-like apical part. Cercus with 13 combs of small spines (retinacula).

Female. Unknown.*Biology*. Unknown.*Etymology*. The species name refers to the Caucasian type locality.

### 3.2. DNA Analysis

DNA analysis was able to further confirm the morphological conclusions. The cytochrome c oxidase subunit 1 (COI) sequence of *D. caucasica* sp. n. was significantly different from other *Docosia* species found in GenBank. Further analysis revealed that while the newly described species was most similar to *D. landrocki*, with which it grouped together on 97% of bootstrap trees (Figure 4), genetic distance between the two was still considerable. In fact, the COI sequence of *D. landrocki* varied form that of *D. caucasica* sp. n. by 9.92% base differences per site (Table 1).

## 4. Discussion

In spite of more than two centuries of research history regarding Mycetophilidae in Europe that could have provided an exhaustive synopsis, the regional fauna still conceals a substantial amount of undescribed diversity (cf. [26]). However, in other regions of the world, Mycetophilidae are even more superficially studied and/or have been almost neglected for a long time, as is the case in Transcaucasia. The first checklist of Mycetophilidae in Georgia was published only recently, still representing probably less than half of the actual local diversity, which in turn undoubtedly includes a substantial number of undescribed species [27]. Therefore, discovery of a new species from the Lesser Caucasus in Georgia is unsurprising.

*Docosia caucasica* sp. n. was represented by a single specimen, which is relatively common as 33% of Georgian species are so far known from singletons [27]. Moreover, in the Malaise trap sample containing the holotype, seven out of the eight additional Mycetophilidae species were also represented by only a single specimen. This bias towards rarity can be explained by insufficient sampling, but as discussed by earlier authors, a substantial proportion of diversity tends to be rare [28,29]. Flather and Sieg [30] considered the majority of species in an ecological community to be represented by a few individuals only. As an example, Coddington et al. [31] described a high frequency of singletons (= 32% on average) in tropical arthropod surveys. In turn, Lim et al. [29] discussed that additional sampling is usually of limited help in decreasing the proportion of the singletons, because with supplementary material even more new singleton species are being collected. For communication purposes, whether inside and/or outside the scientific community, those rare species require naming. Consequently, a considerable proportion of new species have so far been described from only singletons, representing a common practise in entomology. According to Deng et al. [32], from about 4800 new insect species described in *ZooKeys* between 2009 and 2017, more than 20% were based on a single specimen. In the case of Mycetophilidae, 410 new species have been described during the last 20 years (2001–2020) in *Zootaxa*, the world’s foremost journal of zootaxonomy [33], whereas 33% of them are described from a singleton.

Indeed, according to the International Code of Zoological Nomenclature, there are no restrictions to describing new species from singletons until other requirements are met [34,35]. The new species must be unambiguously diagnosed in a way that ensures its identification when recollected. Moreover, deposited physical type material is also mandatory, as that allows re-examination and, if necessary, further testing of the scientific hypothesis regarding species delimitation [36]. In the case of Mycetophilidae, species-specific morphological characters are found mostly in male terminalia, whereas females are frequently determined only to the genus level or not included in a study at all. Therefore, a newly found Mycetophilidae species should be described with emphasize on the morphology of the male terminalia. However, several species have been historically described from a single female holotype, something that has generally been avoided during more recent decades (and should be abstained from in the future). In fact, female-based descriptions usually lead to many subsequent disadvantages, because the association of sexes in Mycetophilidae is frequently questionable and grounded in vague morphological argumentation. However, combining such information with DNA sequence data provides a far more solid basis for unambiguously determining which females and males belong to the same species (e.g. [37]).

The upturn of *Docosia* studies in Europe as well as in other regions has resulted in a number of descriptive and phylogeny based papers [7], forming a solid background for new species discovery and formal description. As shown by Fontaine et al. [38], a recent revision of the group has considerably shortened the time between species collecting and description that was otherwise approximately 21 years. In the case of *Docosia*, it has not been a single monographic revision but a set of papers, which have introduced proper diagnoses, including detailed figures of the male terminalia, for the majority of extant species. Moreover, for at least 30 of the species, genetic information in the form of most frequently used nuclear and mitochondrial markers is now publicly available [7]. Previous studies of Palaearctic *Docosia* species (e.g. [7,9,12,13]) show that the diagnostic characters of male terminalia are conservative, allowing for species identification based on minimal material. Taking into account the rather well described and illustrated assemblage of *Docosia* species as well as the extensive material studied earlier, we are convinced of having delimited a new species—*Docosia caucasica* sp. n.

Based on a combined analysis of five DNA markers, Ševčík et al. [7] discussed intrageneric phylogeny of *Docosia* with distinguishing a group of species having also a well-defined morphology of the male terminalia: longitudinal tergite 9, distinct posterolateral processes of gonocoxites and apically modified setae at posteroventral margin of the gonocoxites medially. In addition to six species, namely *D. nigra*, 1928, *D. landrocki*, *D. anatolica*, *D. diutina*, *D. rohaceki* and *D. pannonica*, included in the group by Ševčík et al. [7], several other species appear to belong here as well, namely *D. helveola* Chandler, 2004, *D. turkmenica* Zaitzev, 2011, *D. blagoderovi* Kurina and Ševčík, 2012, *D. distributa* Kurina and Ševčík, 2012 and *D. trispinosa* Kurina and Ševčík, 2013. Within the group, *D. caucasica* sp. n. is most similar to *D. landrocki*, distinguished by characters as shown in Diagnosis above.

It should be noted, that *D*. *caucasica* sp. n. proved to be distinct from other *Docosia* species based on the commonly used genetic marker cytochrome c oxidase subunit 1 (COI). This was in accordance with the morphological finds, especially as based on further analysis, *D. landrocki* was indeed the closest known relative to the new species. DNA barcoding based on mitochondrial sequences alone, especially when using universal primers, may lead to erroneous conclusions [39]. However, in this case the COI sequence of *D. caucasica* sp. n. fit well into the narrative created by previous phylogenetic studies on *Docosia* [7]. Indeed, according to analysis, the genetic distance between *D. caucasica* sp. n. and *D. landrocki* is comparable to the differences among other known species of this genus (Table 1).

To conclude, in the aforementioned circumstances we consider it better to describe a new species from a single male specimen and keep the wider audience informed, rather than store the material in a drawer and wait decades for further data. In addition, we have not described the new species in isolation, something that is discouraged by multiple taxonomic journals. Instead, it was compared to information from a large set of recent taxonomy papers on Palaearctic *Docosia* and discussed in that context. Nevertheless, high-quality standards in a new species delimitation must be implemented and there cannot be any trade-off on the basis of insufficient material.

## Figures and Tables

**Figure 1 insects-12-01069-f001:**
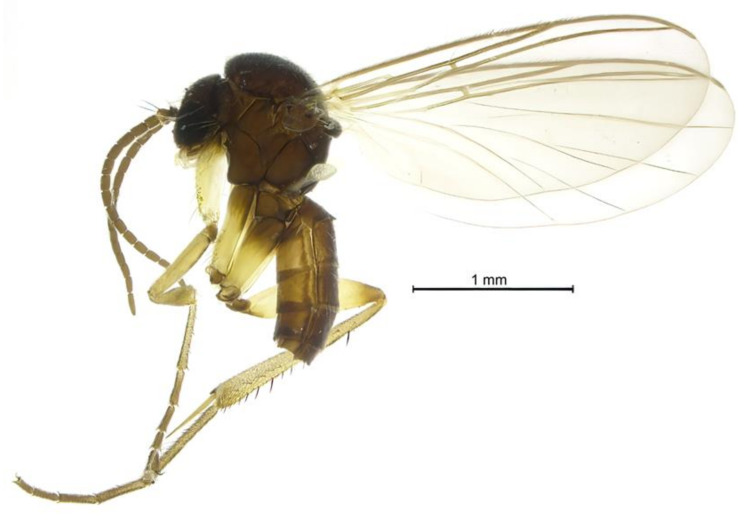
*Docosia caucasica* sp. n., ♂ (ZFMK-DIP-00082491), habitus, terminalia detached.

**Figure 2 insects-12-01069-f002:**
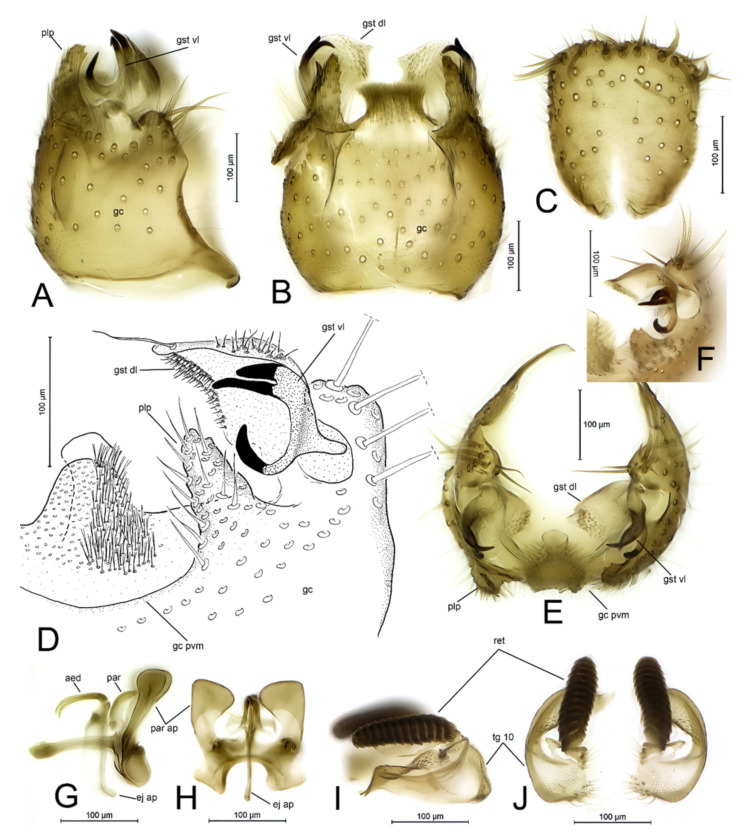
*Docosia caucasica* sp. n., ♂ [ZFMK-DIP-00082491], terminalia. (**A**) Lateral view; (**B**) Ventral view; (**C**) Dorsal view of tergite 9; (**D**) Postero-ventral view of gonostylus and gonocoxite; (**E**) Posterior view; (**F**) Postero-ventral view of gonostylus; (**G**) Lateral view of aedeagal complex; (**H**) Ventral view of aedeagal complex; (**I**) Lateral view of cerci; (**J**) Dorsal view of cerci. Abbreviations: aed = aedeagus; ej ap = ejaculatory apodeme; gc = gonocoxite; gc pvm = posteroventral margin of gonocoxite; gst vl = ventral lobe of gonostylus; gst dl = dorsal lobe of gonostylus; par = paramere; par ap = parameral apodeme; plp = posterolateral process of gonocoxite; ret = combs of retinacula.

**Figure 3 insects-12-01069-f003:**
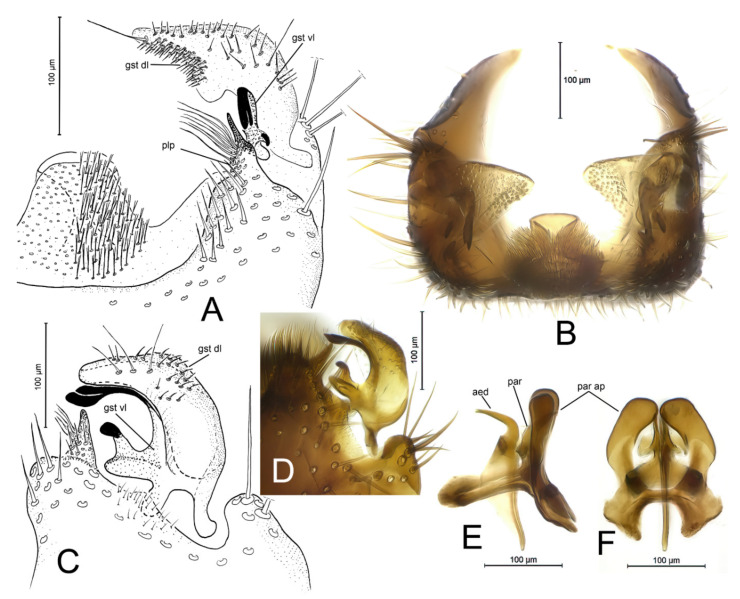
*Docosia landrocki* Laštovka and Ševčík, 2006, ♂ (IZBE0222620), terminalia. (**A**) Postero-ventral view of gonostylus and gonocoxite; (**B**) Posterior view. (**C**,**D**) Lateral view of gonostylus. (**E**) Lateral view of aedeagal complex. (**F**) Ventral view of aedeagal complex. For abbreviations see Figure 2.

**Figure 4 insects-12-01069-f004:**
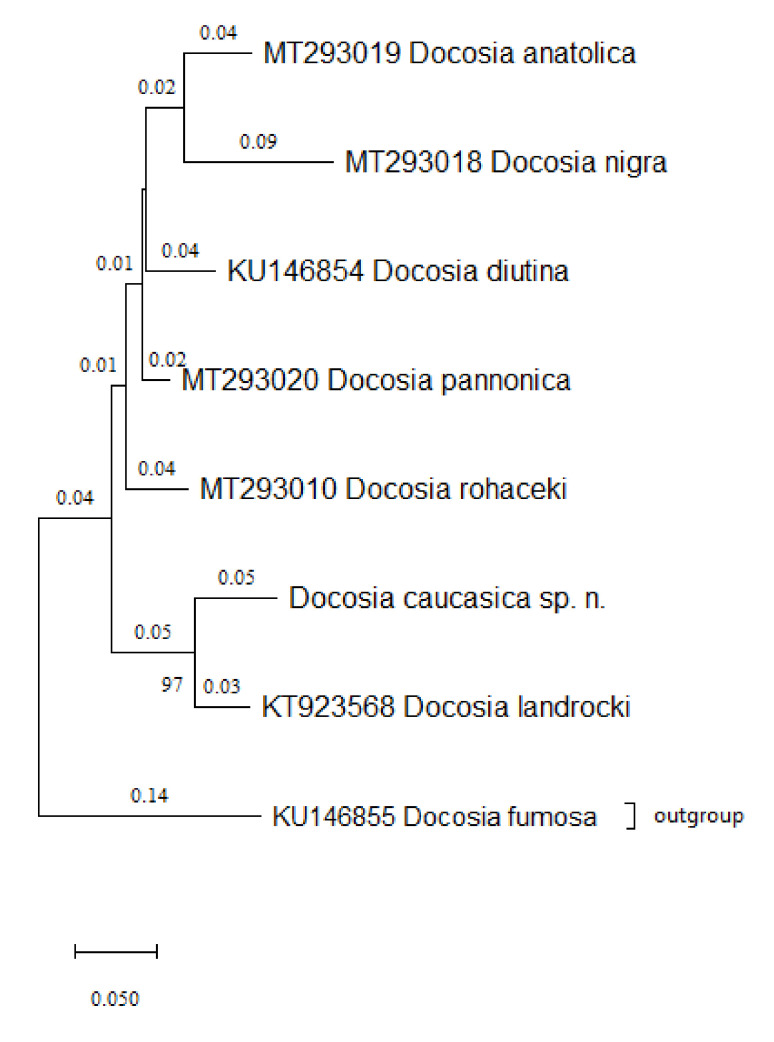
Maximum likelihood hypothesis for the relationships between a group of *Docosia* spp., constructed by using the General Time Reversible model [24]. A discrete Gamma distribution was used to model evolutionary rate differences among sites (5 categories (+G, parameter = 0.2193)), and the tree is drawn to scale.

**Table 1 insects-12-01069-t001:** Table showing the pairwise Kimura 2-parameter distances between sequences [25]. Standard error estimates are shown above the diagonal.

	1.	2.	3.	4.	5.	6.	7.	8.
1. OL619794 *D. caucasica* sp. n.		0.024	0.030	0.036	0.034	0.046	0.049	0.052
2. KT923568 *D. landrocki*	0.099		0.035	0.028	0.038	0.033	0.050	0.045
3. MT293010 *D. rohaceki*	0.140	0.153		0.016	0.024	0.022	0.043	0.041
4. MT293020 *D. pannonica*	0.156	0.124	0.068		0.019	0.016	0.029	0.035
5. MT293019 *D. anatolica*	0.154	0.179	0.116	0.083		0.025	0.038	0.054
6. KU146854 *D. diutina*	0.199	0.155	0.094	0.066	0.112		0.044	0.043
7. MT293018 *D. nigra*	0.208	0.216	0.192	0.135	0.165	0.190		0.047
8. KU146855 *D. fumosa* (outgroup)	0.246	0.216	0.205	0.171	0.261	0.209	0.223	

## Data Availability

All data is available in this paper.

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
