# Peer review of "Every Single Specimen Counts: A New Docosia Winnertz (Diptera: Mycetophilidae) Species Described from a Singleton†"

_insects, 2021, doi:10.3390/insects12121069_

Round 1
Reviewer 1 Report
Nice paper, excellent photos. Some suggestions:
line 156/157 would read more smoothly if it were: ...medially somewhat tapering with a wide lobe...
Current layout has split Table 1 between pages 6 & 7; be better to see Table 1 all on 1 page
line 211 could probably just say: ...tropical arthropod surveys...
line 215. "those rare species are claiming for names likewise" was confusing to me. Maybe "those rare species require naming."?
line 250: "...on minimal material"
line 260: instead of "...other species are evidently assembling,..." how about, "...other species appear to belong here as well,..."
line 263 "distinguished" rather than "distinguishing"
line 281: "must be" rather than "must to be"
I don't agree with the last sentence in the Discussion...it seems to imply that describing a new species based on a singleton, but without support from DNA is inadequate. In many insect groups male terminalia are the standard for species delimitation and I don't think it is useful to suggest that this character system is, by itself, not enough. Unlike Docosia, most insect groups do not have a large number of species sequenced, so in most other insect groups sequencing of a new species would not provide additional useful information. Furthermore, many morphologists do not have access to or skills to perform molecular methods. All these things considered, I believe it would be a hinderance to the science if DNA were a new requirement of species description (from singletons or otherwise) because it would ultimately slow down or even prevent species descriptions. In the case of the current paper, the DNA is a nice addition because it allowed you to place the species into a previously established molecular context, but I think it would have been equally fine to describe the new species based on morphology with a discussion of relatedness based solely on morphological traits.
Author Response
Response to the reviewer’s comments (Reviewer 1)
We would like to express our thanks to the Reviewer for his/her very instructive and profound comments. Thank you very much for your time. All suggested improvements are considered.
1) line 156/157 would read more smoothly if it were: ...medially somewhat tapering with a wide lobe...
Reply: Agree and changes made.
2) Current layout has split Table 1 between pages 6 & 7; be better to see Table 1 all on 1 page
Reply: Should be on one page now. Actually, on my screen it has been always on one page.
3) line 211 could probably just say: ...tropical arthropod surveys...
Reply: Agree and changes made.
4) line 215. "those rare species are claiming for names likewise" was confusing to me. Maybe "those rare species require naming."?
Reply: Agree and changes made.
5) line 250: "...on minimal material"
Reply: Agree and changes made.
6) line 260: instead of "...other species are evidently assembling,..." how about, "...other species appear to belong here as well,..."
Reply: Agree and changes made.
7) line 263 "distinguished" rather than "distinguishing"
Reply: Agree and changes made.
8) line 281: "must be" rather than "must to be"
Reply: Agree and changes made.
9) I don't agree with the last sentence in the Discussion...it seems to imply that describing a new species based on a singleton, but without support from DNA is inadequate. In many insect groups male terminalia are the standard for species delimitation and I don't think it is useful to suggest that this character system is, by itself, not enough. Unlike Docosia, most insect groups do not have a large number of species sequenced, so in most other insect groups sequencing of a new species would not provide additional useful information. Furthermore, many morphologists do not have access to or skills to perform molecular methods. All these things considered, I believe it would be a hinderance to the science if DNA were a new requirement of species description (from singletons or otherwise) because it would ultimately slow down or even prevent species descriptions. In the case of the current paper, the DNA is a nice addition because it allowed you to place the species into a previously established molecular context, but I think it would have been equally fine to describe the new species based on morphology with a discussion of relatedness based solely on morphological traits.
Reply: Agree, the last sentence is left out as redundant.
Reviewer 2 Report
The manuscript titled "Every single specimen can “rock”: a new Docosia Winnertz (Diptera: Mycetophilidae) species described from singleton" provides a new species description of an interesting mycetophilid Diptera. Manuscript is generally well written and I have no major concerns about it. Since it is only an one new species description with some remarks, I would consider treating it as "Correspondence" and not as "Article". The taxonomic part is very nicely done. There are some minot issues which need to be resolved (see further comments). Overall, however, I find this manuscript of interest to readers and worth to be published in Insects as "Correspondence". Below are some suggestions and comments to improve the manuscript:
Title
I think the "rock" word is little bit overblown, and so I suggest the following title: Every single specimen counts: a new Docosia Winnertz (Diptera: Mycetophilidae) species described from a singleton." (article added before singleton)
Introduction
lines 37,38 - Regions/region should be modified to "Realms/Realm" (as it is more precise term)
Material and methods
You should list which species you compared with the new species and from which museums. The last sentence here indicates you compared only D. landrocki?
lines 73 and 74 - author names and years of description should be included in each genus or species when it appears for the first time in manuscript. Please add them (they are given in Discussion, but they should be used here, because here it is for the first time).
Regarding phylogenetic analysis, I really miss information about analysis (ML, Bayesian?) in this section. Something is written in the figure caption near the tree but it must be here. Phylogenetic analysis is not described at all (which models, settings,..?)
Results
line 177 - consider replacing "finds" with "conclusions"
lines 182-183 - please use percentage for this (0.062)
Figure 4 - caption should be less informative; instead, please add all info about analysis, model selection, settings, etc to Material and methods section
Why the tree was rooted with the species of the same genus as ingroup? Usually a member of some other genus is treated as an outgroup.
Table 1 - please replace commas with dots
Table 1 - xxxxxxxxxx looks weird, please use just "-" or something like this
Discussion
First I thought that the discussion on the description of a species from a singleton is redundant but after reading it again I changed my mind and I think it might be quite interesting. Although some parts of Discussion are a bit lenghty and can be shortened, geenrally it is well written.
lines 257-262 - author and year after each taxon should be added only when it appears fro the first time in ms
line 281 - "must to" delete "to"
line 283 - add "e.g." between like and DNA
Author Response
Response to the reviewer’s comments (Reviewer 2)
We would like to express our thanks to the Reviewer for his/her very instructive and profound comments. Thank you very much for your time. Most of the suggested changes are considered, otherwise, they are argued below.
1) The manuscript titled "Every single specimen can “rock”: a new Docosia Winnertz (Diptera: Mycetophilidae) species described from singleton" provides a new species description of an interesting mycetophilid Diptera. Manuscript is generally well written and I have no major concerns about it. Since it is only an one new species description with some remarks, I would consider treating it as "Correspondence" and not as "Article". The taxonomic part is very nicely done. There are some minor issues which need to be resolved (see further comments). Overall, however, I find this manuscript of interest to readers and worth to be published in Insects as "Correspondence". Below are some suggestions and comments to improve the manuscript:
Reply: We will leave this decision for the editor(s). However, 10 pages are probably too much for a “Correspondence”
2) I think the "rock" word is little bit overblown, and so I suggest the following title: Every single specimen counts: a new Docosia Winnertz (Diptera: Mycetophilidae) species described from a singleton." (article added before singleton)
Reply: Thank you very much for your suggestion that is undoubtedly a more conventional and polite redaction. However, we have used the “rock” intentionally as an eye-catching and somewhat aggressive expression to attract a wider audience, i.e. not only fungus gnats’ students. Moreover, two other reviewers have found the title fully acceptable. We added an article before singleton as you suggested.
However, if you still find that change unavoidable, we will be at your disposal.
3) lines 37,38 - Regions/region should be modified to "Realms/Realm" (as it is more precise term)
Reply: Agree and changes made.
4) You should list which species you compared with the new species and from which museums. The last sentence here indicates you compared only D. landrocki?
Reply: Agree and changes made. All the comparative material is stored in IZBE. D. landrocki is highlighted here as the closest species to D. caucasica sp. n.
5) lines 73 and 74 - author names and years of description should be included in each genus or species when it appears for the first time in manuscript. Please add them (they are given in Discussion, but they should be used here, because here it is for the first time).
Reply: Agree and changes made.
6) Regarding phylogenetic analysis, I really miss information about analysis (ML, Bayesian?) in this section. Something is written in the figure caption near the tree but it must be here. Phylogenetic analysis is not described at all (which models, settings,..?)
Reply: Agree and changes made.
7) line 177 - consider replacing "finds" with "conclusions"
Reply: Agree and changes made.
8) lines 182-183 - please use percentage for this (0.062)
Reply: Agree and changes made.
9) Figure 4 - caption should be less informative; instead, please add all info about analysis, model selection, settings, etc to Material and methods section
Reply: Agree and changes made.
10) Why the tree was rooted with the species of the same genus as ingroup? Usually a member of some other genus is treated as an outgroup.
Reply: D. fumosa was chosen as the outgroup based on previous work done by Ševčík et al. in 2020, where they used sequence data from five markers to create a phylogenic tree for selected species of Docosia. On that figure, D. fumosa is relatively distant from the cluster containing the other species used in our study. Furthermore, the tree branches have strong bootstrap values in their article. Therefore, we are confident that D. fumosa does not belong to the ingroup of our study. However, choosing a species from another genus for the outgroup is of course absolutely valid. If our explanation has failed to satisfy your concerns, we would be happy to change the outgroup species.
11) Table 1 - please replace commas with dots
Reply: Agree and changes made.
12) Table 1 - xxxxxxxxxx looks weird, please use just "-" or something like this
Reply: This is the GeneBank sequence number of the new species that will be provided upon acceptance of the ms.
13) Discussion
First I thought that the discussion on the description of a species from a singleton is redundant but after reading it again I changed my mind and I think it might be quite interesting. Although some parts of Discussion are a bit lenghty and can be shortened, geenrally it is well written.
lines 257-262 - author and year after each taxon should be added only when it appears fro the first time in ms
Reply: Agree and changes made.
14) line 281 - "must to" delete "to"
Reply: Agree and changes made.
15) line 283 - add "e.g." between like and DNA
Reply: Following the suggestions of the first reviewer, the last sentence is left out as redundant.
Reviewer 3 Report
Dear Authors,
The manuscript entitled as “Every single specimen can “rock”: a new Docosia Winnertz (Diptera: Mycetophilidae) species described from singleton” is very well written and illustrated.
The new species is described in a detailed manner and the associated photos and illustrations represent a high quality.
In addition to the alphataxonomic treatise, authors also provide DNA barcoding, phylogenetic analysis and discussion about singletons. These all aspects strengthen the ms and are valuable.
The phylogenetic analysis is perhaps too heavy procedure, given that only one gene is being studied. It would even be very fine with a simple NJ-tree and K2P distances.
In fact, because BOLD uses K2P distances and most diptera taxonomists are not experts in advanced genetic methods, I encourage that you add K2P distances also (e.g. Table 2 similar to Table 1). At least I would then better grasp how distant your new species is e.g. to D. landrocki.
Author Response
Response to the reviewer’s comments (Reviewer 3)
We would like to express our thanks to the Reviewer for his/her very instructive and profound comments. Thank you very much for your time. Your suggestion is considered as described below.
1) In fact, because BOLD uses K2P distances and most diptera taxonomists are not experts in advanced genetic methods, I encourage that you add K2P distances also (e.g. Table 2 similar to Table 1). At least I would then better grasp how distant your new species is e.g. to D. landrocki.
Reply: Agree and changes made. Table 1 is showing the pairwise Kimura 2-parameter distances now.